# Artificially inserted strong promoter containing multiple G-quadruplexes induces long-range chromatin modification

**Shuvra Shekhar Roy[1,2], Sulochana Bagri[1,2], Soujanya Vinayagamurthy[1,2], Avik Sengupta[3], Claudia Regina Then[4], Rahul Kumar[3], Sriram Sridharan[4], Shantanu Chowdhury[1,2]\***

[1]CSIR-Institute of Genomics and Integrative Biology, New Delhi, India; [2]Academy of Scientific and Innovative Research (AcSIR), Ghaziabad, India; [3]Department of Biotechnology, Indian Institute of Technology Hyderabad, Hyderabad, India; [4]Cancer Science Institute of Singapore, National University of Singapore, Singapore, Singapore

**\*For correspondence:**
shantanuc@igib.in

**Competing interest:** The authors declare that no competing interests exist.

**Abstract** Although the role of G-quadruplex (G4) DNA structures has been suggested in chromosomal looping this was not tested directly. Here, to test causal function, an array of G4s, or control sequence that does not form G4s, were inserted within chromatin in cells. In vivo G4 formation of the inserted G4 sequence array, and not the control sequence, was confirmed using G4-selective antibody. Compared to the control insert, we observed a remarkable increase in the number of 3D chromatin looping interactions from the inserted G4 array. This was evident within the immediate topologically associated domain (TAD) and throughout the genome. Locally, recruitment of enhancer histone marks and the transcriptional coactivator p300/Acetylated-p300 increased in the G4-array, but not in the control insertion. Resulting promoter-enhancer interactions and gene activation were clear up to 5 Mb away from the insertion site. Together, these show the causal role of G4s in enhancer function and long-range chromatin interactions. Mechanisms of 3D topology are primarily based on DNA-bound architectural proteins that induce/stabilize long-range interactions. Involvement of the underlying intrinsic DNA sequence/structure in 3D looping shown here therefore throws new light on how long-range chromosomal interactions might be induced or maintained.

## eLife assessment

This **valuable** study demonstrates that genomic insertion of a G4-containing sequence can be sufficient to induce chromosome loops and alter gene expression. The evidence supporting the conclusions is **convincing**. Effects were shown by Hi-C as well as qPCR for chromatin modifications and expression, and the specificity of the effects was controlled by mutating the G4-containing sequence or treating with LNA probes to abolish G4 structure formation. The work will be of interest to researchers working on chromatin organization and gene regulation.

## Introduction

G-quadruplexes (G4s), non-canonical DNA secondary structures with quartets of Guanines bonded by Hoogsteen base pairing, are instrumental in regulating gene expression (*Sengupta et al., 2021*; *Varshney et al., 2020*). G4s were primarily observed to be able to regulate gene expression when

present around transcription start sites (TSSs; *Huppert and Balasubramanian, 2007*; *Rawal et al., 2006*; *Verma et al., 2008*). G4s can regulate gene expression by directly regulating recruitment of transcription factors and RNA polymerase or via alteration of DNA accessibility by modulating the epigenetic state of the gene promoters (*Hussain et al., 2017*; *Kumar et al., 2011*; *Lago et al., 2021*; *Mukherjee et al., 2019*; *Saha et al., 2017*; *Sharma et al., 2021*; *Varshney et al., 2020*). Recent studies have implicated the role of G4s in long-distance gene regulation (*Robinson et al., 2021*).

High-throughput chromosome conformation capture techniques reveal that specific regions of the human genome interact in three dimensions (3D) via chromatin looping and formation of topologically associated domains (TADs; *Bonev and Cavalli, 2016*; *Denker and de Laat, 2016*; *Roy et al., 2018*). Interestingly, recent computational studies observed enrichment of G4s in TAD boundaries along with higher enrichment of architectural proteins like CTCF and cohesin (*Hou et al., 2019*). Further, multiple studies noted the presence of G4s correlated with enhancer histone marks like H3K27Ac and H3K4Me1, and predominantly open chromatin regions (*Calo and Wysocka, 2013*; *Hou et al., 2021*; *Shlyueva et al., 2014*).

Although these studies implicate the role of G4s in long-range interactions and/or enhancer function, this was not directly tested. Here we asked if G4 structures might directly alter 3D chromatin, and affect long-range interactions including the epigenetic state of chromatin. To address this, we inserted an array of G4s into an isolated locus devoid of G4-forming sequences using CRISPR-Cas9 genome editing. To evaluate the specific function of G4s, a similar sequence of identical length but devoid of G4-forming capability was introduced. Using these pair of cell lines, we observed insertion of G4s specifically led to the recruitment of enhancer histone marks and increased expression of genes in a 10 Mb window. 3 C and Hi-C results showed induced long-range interactions throughout the genome affecting topologically associated domains (TADs) that were specifically due to the incorporated G4s, and not found in case of the control insertion.

## Results

### Insertion of an array of G4s in an isolated locus

First, we sought to insert an array of G4s in a relatively isolated locus. We looked into Hi-C data from *Rao et al., 2014* and identified a region that was markedly isolated with little or no interaction with its surrounding regions (as shown by snapshots of Hi-C interaction matrices obtained using the 3D genome browser *Wang et al., 2018* in *Figure 1—figure supplement 1*). In addition, this region was devoid of any G4s in the vicinity (no G4 forming motifs in a±2.5 kb window). Thereafter, we artificially inserted an array of G4 forming sequences (275 bp long) at this region near the 79 millionth position of chromosome 12 (79M in following text, chr12:79872423–79872424, hg19 genome assembly) using CRISPR-Cas9 genome editing (*Figure 1A*, *Figure 1—figure supplement 2*). To study specific effects due to G4s, if any, a control sequence of identical length was inserted in HEK293T cells at the same locus where specific G/Cs necessary for G4 formation were substituted so that G4s are not formed by this sequence (G4-mutated control, *Figure 1A*, *Figure 1—figure supplement 2*); we also ensured that the GC content was minimally affected by the substitutions (72.4% from 76.73%). Homozygous insertion was confirmed by PCR using primers adjacent to the insertion site followed by Sanger sequencing (*Figure 1B*, *Figure 1—figure supplement 3*). The array of G4-forming sequences used for insertion was previously reported to form stable G4s in human cells (*Lim et al., 2010*; *Monsen et al., 2020*; *Palumbo et al., 2009*; *Sharma et al., 2021*).

### Chromatin epigenetic landscape upon insertion of G4s

To understand how the formation of G4s altered the local chromatin, chromatin immunoprecipitation (ChIP) of different chromatin-modifying histone marks was done followed by qRT-PCR using primers spanning the inserted locus. PCR primers were designed such that none of the primers bind to any site of G/C alteration in the mutated control insert; either the forward/reverse primer is from the adjacent region for specificity; covers adjacent regions for studying any effects on chromatin; and, PCRs optimized keeping in mind the repeats within the inserted sequence. Given these, primer pairs R1-R4 were chosen for further work following optimizations (*Figure 2*, top panel). For G4 formation within cells by the G4-array insert sequence we used the reported G4 antibody BG4 (*Hänsel-Hertsch et al., 2016*). Using primer pairs R2, covering >100 bases of the inserted G4-array, or the G4-mutated

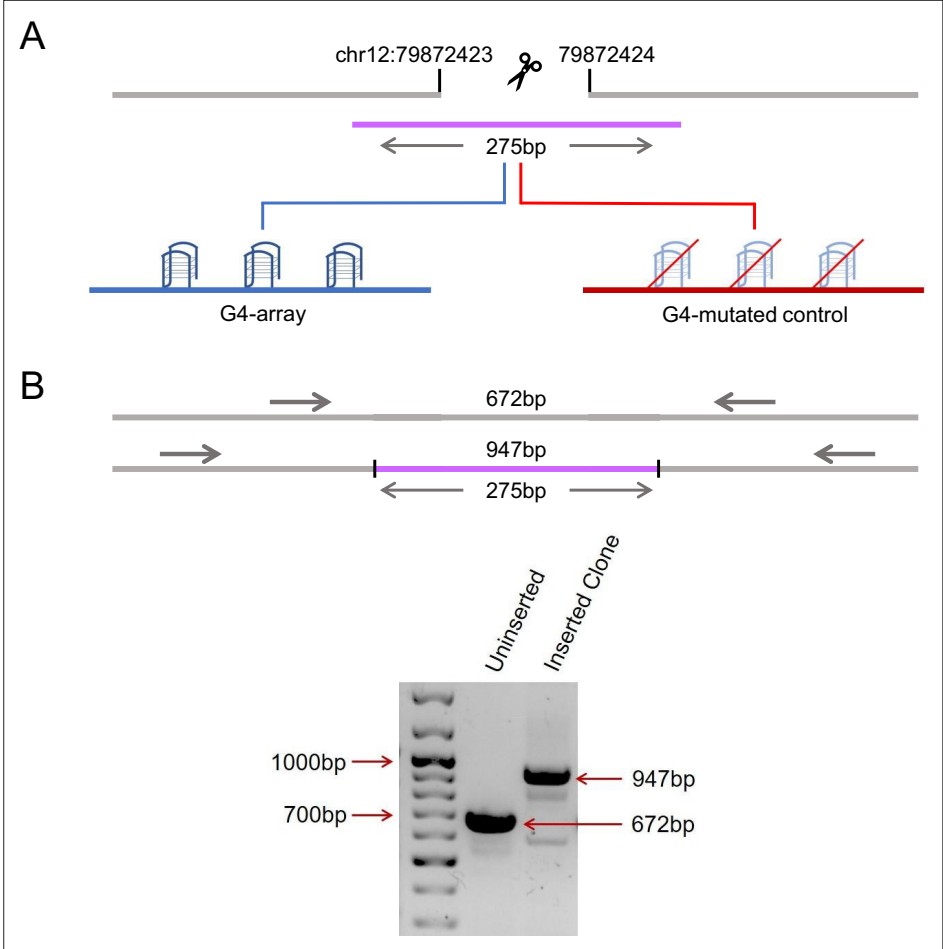

**Figure 1.** Insertion of an array of G4s in an isolated locus. (**A**) Schematic showing the insertion of the G4-array and the G4-mutated control at chr12:79,872,423–79,872,424 (hg19). (**B**) PCR of the insertion locus showing the successful insertion of the 275 bp long insert sequence.

The online version of this article includes the following source data and figure supplement(s) for figure 1:

**Source data 1.** Raw blot shown in *Figure 1B*.

**Figure supplement 1.** An isolated locus was chosen for insertion.

**Figure supplement 2.** Insert sequences.

**Figure supplement 3.** Insertion confirmed by Sanger sequencing.

---

control, BG4 ChIP followed by qPCR was performed. Significant BG4 binding was clear in the G4-array insert, and not in the G4-mutated insert, demonstrating formation of G4s by the inserted G4-array (*Figure 2—figure supplement 1*).

We observed significant increase in H3K4Me1 and H3K27Ac enhancer marks in the G4-array when compared to the G4-mutated control (*Figure 2A and B*). However, there was no G4-specific change in the presence of chromatin compaction marks, H3K27Me3 and H3K9Me3, or the promoter activation mark H3K4Me3 (*Figure 2C, D and E*). The G4-dependent recruitment of H3K4Me1 (associated with enhancers *Heintzman et al., 2009*; *Heintzman et al., 2007*) and H3K27Ac (associated with active enhancers and promoters *Creyghton et al., 2010*; *Heintzman et al., 2009*; *Heintzman et al., 2007*) indicated enhancer-like characteristics of the inserted G4s.

## Enhancer-like features emerged upon insertion of G4s

We next asked how the insertion of the G4-array influenced the expression of surrounding genes. To understand the distance-dependent gene regulatory impacts of the inserted G4-array, the mRNA expression of the nearest three genes and then some arbitrarily chosen genes further away up to 5

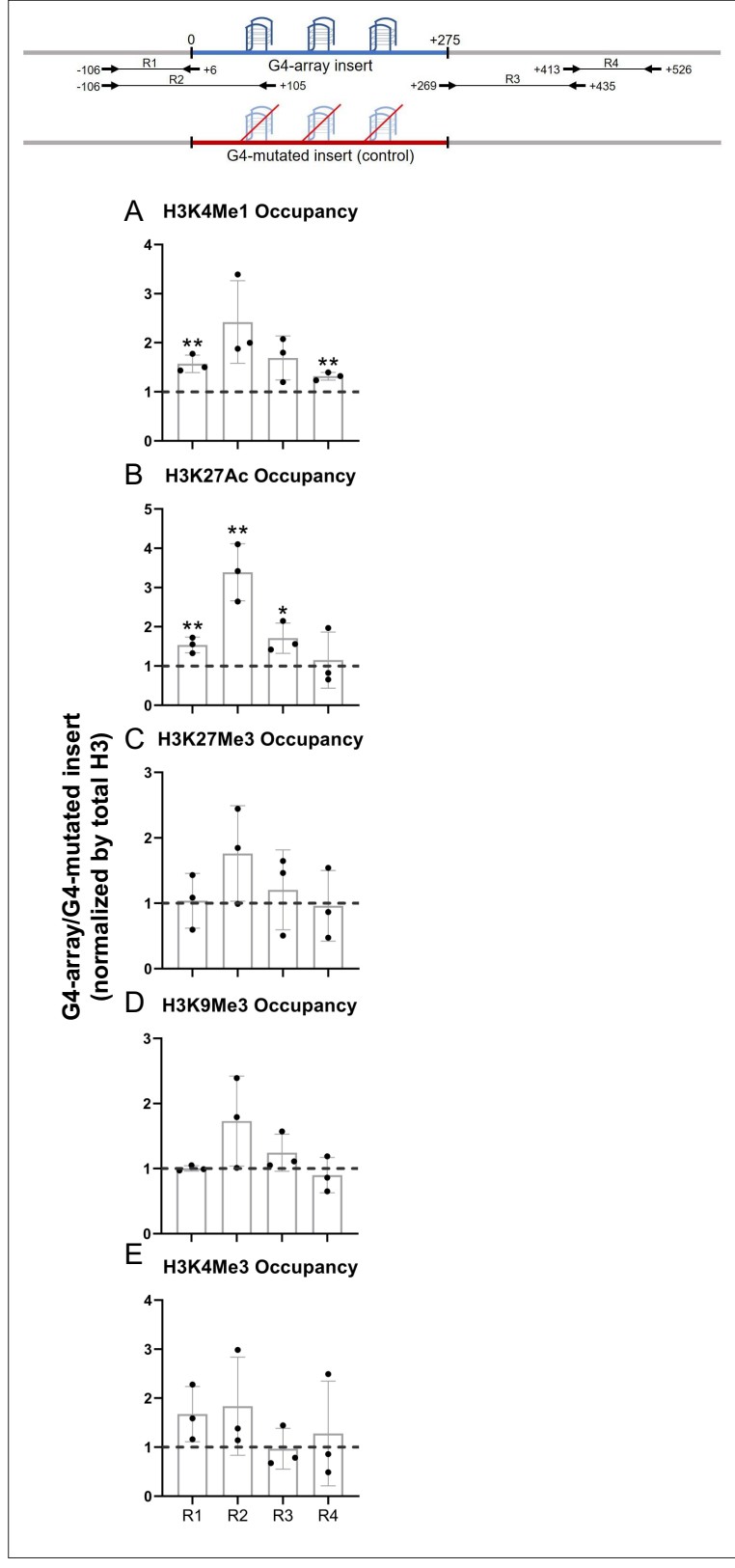

**Figure 2.** Changes in chromatin upon G4-array insertion. The top panel shows the positions of the PCR amplicons used in the Histone ChIP experiments. Changes in chromatin-modifying histone modifications in the insert region represented by calculating the ratio of occupancy of different histone marks in the G4-array insert cells over the G4-mutated insert (control) cells- enhancer mark, H3K4Me1 (**A**); active enhancer/promoter mark, H3K27Ac

*Figure 2 continued on next page*

*Figure 2 continued*

(**B**); facultative repressor mark, H3K27Me3 (**C**); constitutive repressor mark, H3K9Me3 (**D**) and active promoter mark, H3K4Me3 (**E**). Mean ± SD (n=3); unpaired, two-tailed *t*-test (\*p<0.05, \*\*p<0.01, \*\*\*p<0.001, \*\*\*\*p<0.0001).

The online version of this article includes the following source data and figure supplement(s) for figure 2:

**Source data 1.** Source data for *Figure 2* (Changes in chromatin upon G4-array insertion).

**Figure supplement 1.** G4 formation analyzed by BG4 ChIP.

**Figure supplement 1—source data 1.** Source data for *Figure 2—figure supplement 1* (G4 formation analyzed by BG4 ChIP).

---

megabases (Mb) both up and downstream from the insertion site was quantified. Notably, the expression of four of the tested genes (*PAWR, PPP1R12A, NAV3,* and *SLC6A15*) increased in the G4-array insert compared to the mutated insert control cells (*Figure 3A*). Based on this enhanced expression, we further tested and observed a somewhat concomitant increase in the recruitment of Ser5 phosphorylated RNA Pol II in the surrounding gene promoters (*Figure 3B*). Next, we tested if chromosomal looping interactions between the insertion site and the gene promoters were involved in these long-distance effects by using chromosome conformation capture (3C). The 3C assay between the insertion locus and the gene promoters could only be performed till the *NAV3* promoter 1.6 Mb away. Beyond this distance, there was not any significantly detectable PCR amplification of 3C interaction products. The 3C assays revealed that there was a G4-dependent increase in chromosomal looping interactions of the insertion locus with the gene promoters (*Figure 3C*). These results suggested that the inserted G4-array sequence was acting like an enhancer element.

To understand the mechanism behind the enhancer-like property of the inserted G4-array we analyzed the recruitment of transcriptional coactivator p300 (*Kalkhoven, 2004*). There was a relatively modest increase in the recruitment of p300, and a more substantial increase in the recruitment of the more functionally active acetylated p300/CBP, was seen within the G4-array when compared against the mutated control (*Figure 3D and E*). Together, these results supported the enhancer-like function of the inserted G4-array.

## LNA-mediated disruption of the inserted G4s reverses enhancer phenotype

To further establish that the enhancer effects upon the G4 array insertion are due to the formation of G4s, we wanted to see if some of the effects observed could be reversed upon disrupting the inserted structures. Specific Locked Nucleic Acid (LNA) probes were designed to target and disrupt the G4 using a similar approach as shown by others (*Cadoni et al., 2021*; *Chowdhury et al., 2022*; *Kumar et al., 2008*). Three probes were designed with stretches of mostly cytosines (Cs) as LNAs which would hybridize with stretches of guanines (Gs) in the G4-array insert important for the structure formation (*Figure 4A*; see Methods). We observed that there was a significant decrease in the expression of *PPP1R12A* and *NAV3*, two of the genes initially observed to have G4-dependent enhanced expression (*Figure 3A*), when the G4 array inserted cells were treated with the G4 targeting LNAs (*Figure 4B*). As expected, although modest, a decrease in the H3K4Me1 and H3K27Ac enhancer histone modifications was evident within the insert upon the LNAs treatment (*Figure 4C and D*). As a control experiment, we next tested whether the LNA probes affected surrounding gene expression in the G4-mutated insert cells. Changes in the expression of the genes were not significant across replicates in case of G4-mutated insert cells (*Figure 4—figure supplement 1*). Together these confirmed the decrease in the expression of *PPP1R12A* and *NAV3* in the G4-array insert upon LNA treatment was likely specific to G4 disruption. These indicate that the disruption of the inserted G4s can reverse the enhancer functions observed upon G4 insertion, further supporting the role of the G4 structure in enhancer functions.

## Domain-wide increase in looping interactions by G4s

For in-depth analysis of the long-range changes in chromatin architecture upon G4 insertion, we performed genome-wide interaction by Hi-C. First, we compared all the Hi-C contacts originating within a±10 kb window comprising the G4-array insert, or the G4-mutated control insert. Compared to the mutated control, the G4-inserted locus had more than twice as many genome-wide Hi-C

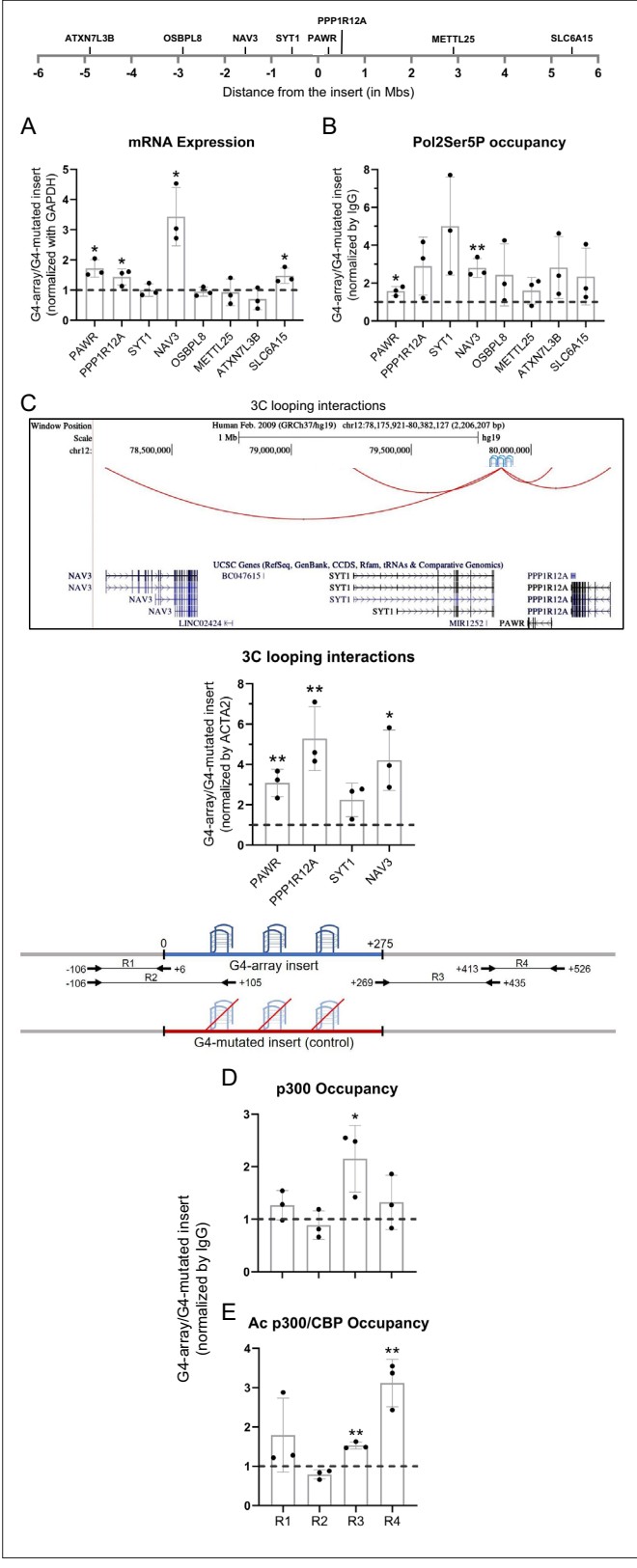

**Figure 3.** Insertion of the G4-array led to enhancer function. (**A**) Long-range G4-dependent changes in mRNA expression are represented by calculating the ratio of expression of surrounding genes in the G4-array insert cells over the G4-mutated insert (control) cells. Top panel shows the positions of the gene promoters with respect to the insertion site. (**B**) Ratio of Pol2 Phospho-Ser5 Occupancy at the promoters of the surrounding genes in the

*Figure 3 continued on next page*

*Figure 3 continued*

G4-array insert cells over the G4-mutated insert (control) cells. (**C**) Fold change in 3C looping interactions between the insertion and the surrounding gene promoters in the G4-array insert cells over the G4-mutated insert (control) cells. The UCSC genome browser snapshot above shows the 3C looping interactions between the insertion and the surrounding gene promoters. The ratio of occupancy of p300 (**D**) and Ac p300/CBP (**E**) in the G4-array insert cells over the G4-mutated insert (control) cells. The panel above shows the positions of the PCR amplicons used in the ChIP experiments. Mean ± SD (n=3); unpaired, two-tailed *t*-test (*p<0.05, **p<0.01, ***p<0.001, ****p<0.0001).

The online version of this article includes the following source data and figure supplement(s) for figure 3:

**Source data 1.** Source data for *Figure 3* (Insertion of the G4-array led to enhancer function).

**Figure supplement 1.** pG4s in the activated gene promoters.

interactions (6390 vs 3133; *Figure 5A and B*, *Supplementary file 1*). To rule out the possibility of artifacts due to the insertion we independently analyzed Hi-C data in HEK293T cells reported earlier (taken from GSE44267, *Zuin and Dixon, 2014*). After normalizing for sequencing depth, the number of Hi-C contacts from the same window in HEK293T was relatively similar to the G4-mutated insert control (3968 and 3133 respectively, *Figure 5C*, *Supplementary file 1*). Together, these showed that a significant number of new long-range interactions were induced throughout the genome due to the inserted G4s, but not from the inserted control sequence.

For closer analysis, we focused on intrachromosomal Hi-C interaction matrices of the G4-array insert, or the mutated control insert. This was centered on the insertion locus on chromosome 12 (chr12:7,80,72,423–8,16,72,423; insertion site marked with arrows in *Figure 6A and B*). The number of Hi-C interactions in the G4-array insert was clearly enriched compared to the G4-mutated insert control, as expected from the global Hi-C contacts noted above. We noted that while the interactions from the G4-array insert were significantly more, the insertion per se did not affect the overall domain architecture, which was largely similar between G4 or G4-mutated inserts as clear from *Figure 6A and B*. Further, we asked if the domain architecture was retained from that seen in HEK293T cells (with no insertion): Comparison using reported HiC data for the same region from HEK293T cells showed this to be the case confirming that the chromatin domain architecture remained relatively unchanged on introducing the G-array or G4-mutated regions (*Figure 6—figure supplement 1*).

To evaluate the effect of G4s in more detail, we plotted a Hi-C heatmap to show the enhanced or reduced (differential) contacts in the G4-array insert compared to the G4-mutated insert control cells (*Figure 6C*; relatively enriched/reduced contacts in the G4-array insert w.r.t. the G4-mutated insert plotted in red or blue, respectively; using Juicebox for analysis). This clearly showed that the G4-array induced significantly more Hi-C interactions; interestingly this was particularly evident in the downstream regions. For a closer analysis, we mapped the interaction frequency in a±100 kb window centered on the insertion site. This clearly showed the difference in the number of interactions between the upstream region vis-a-vis the region downstream of the insertion (*Figure 6D*).

To further confirm we used an independent HiC analysis method, HOMER (Hypergeometric Optimization of Motif EnRichment, *Heinz et al., 2018*) to compute the enhanced/reduced long-range interactions in the G4-array insert, compared to the control G4-mutated insert. Differential analysis using HOMER showed that the inserted locus induced significantly higher number of interactions in the case of G4-array insert relative to the control G4-mutated case (*Figure 6E*). When we plotted the significantly different chromosomal interactions with minimum 20 interaction reads, it was again clear that the number of interactions with the G4-array insertion region was significantly enhanced in the downstream region relative to the upstream (*Figure 6F*).

Together these show a clear role of G4s in inducing long-range interactions. A similar sequence devoid of G4-forming capability did not induce such interactions. Furthermore, the overall nature of the TAD was not disturbed, and largely consistent with what is noted in cells with no insertion. Overall, these support that the insertion of G4s induced long-range interactions with minimal organizational changes in the 3D chromatin domain, underlining the molecular role of G4s in the arrangement of 3D chromatin.

A second significant feature was notable at the insertion locus. The number of induced long-range interactions was more significant downstream of the insertion site, compared to the upstream region (*Figure 6C, D and F*). A close look at the Hi-C contact matrices indicated that the site of insertion was very close and downstream to the TAD boundary (*Figure 6A–C*). We reasoned that the G4-dependent

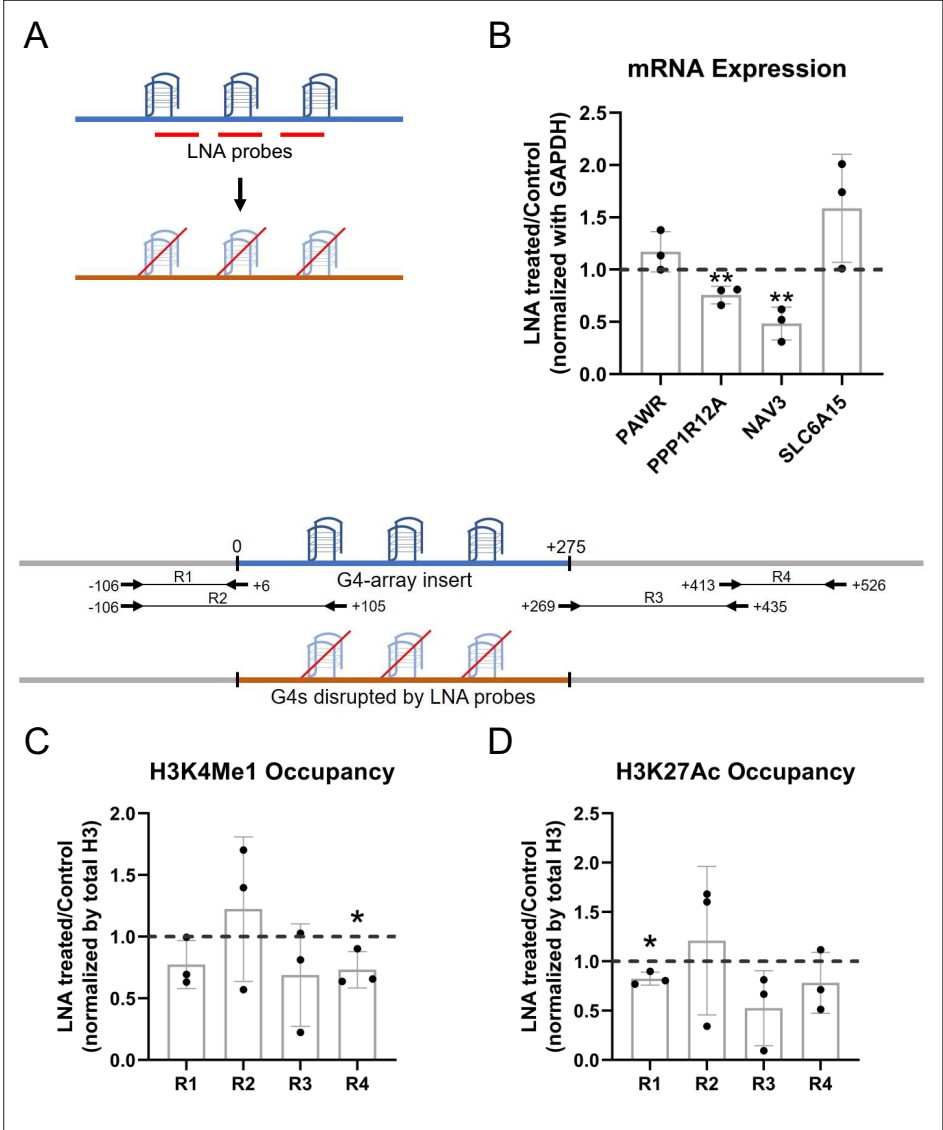

**Figure 4.** LNA-mediated disruption of the inserted G4s reverses enhancer phenotype. (**A**) Schematic showing the inserted G4 structures disrupted using LNA probes (details in methods). (**B**) Effects of LNA treatment in the G4-array insert cells on the expression of surrounding genes which showed enhanced expression when compared against the G4-mutated insert (control) cells in *Figure 3A*- represented by the ratio of expression of surrounding genes in the LNA-treated over the vehicle-treated (control) cells. Effects of LNA treatment in the G4-array insert cells on the levels of H3K4Me1 (**C**) and H3K27Ac (**D**) histone modifications at the insert locus represented by the ratio of occupancy of the histone marks in the LNA-treated over the vehicle-treated (control) cells. The top panel shows the positions of the PCR amplicons used in the Histone ChIP experiments. Mean ± SD (n=3); unpaired, two-tailed t-test (*p<0.05, **p<0.01, ***p<0.001, ****p<0.0001).

The online version of this article includes the following source data and figure supplement(s) for figure 4:

**Source data 1.** Source data for *Figure 4* (LNA-mediated disruption of the inserted G4s reverses enhancer phenotype).

**Figure supplement 1.** Effects of LNA treatment in the G4-mutated insert (control) cells on the expression of surrounding genes represented by the ratio of expression of surrounding genes in the LNA-treated over the vehicle-treated (control) cells.

**Figure supplement 1—source data 1.** Source data for *Figure 4—figure supplement 1* (Effects of LNA treatment in the G4-mutated insert (control) cells).

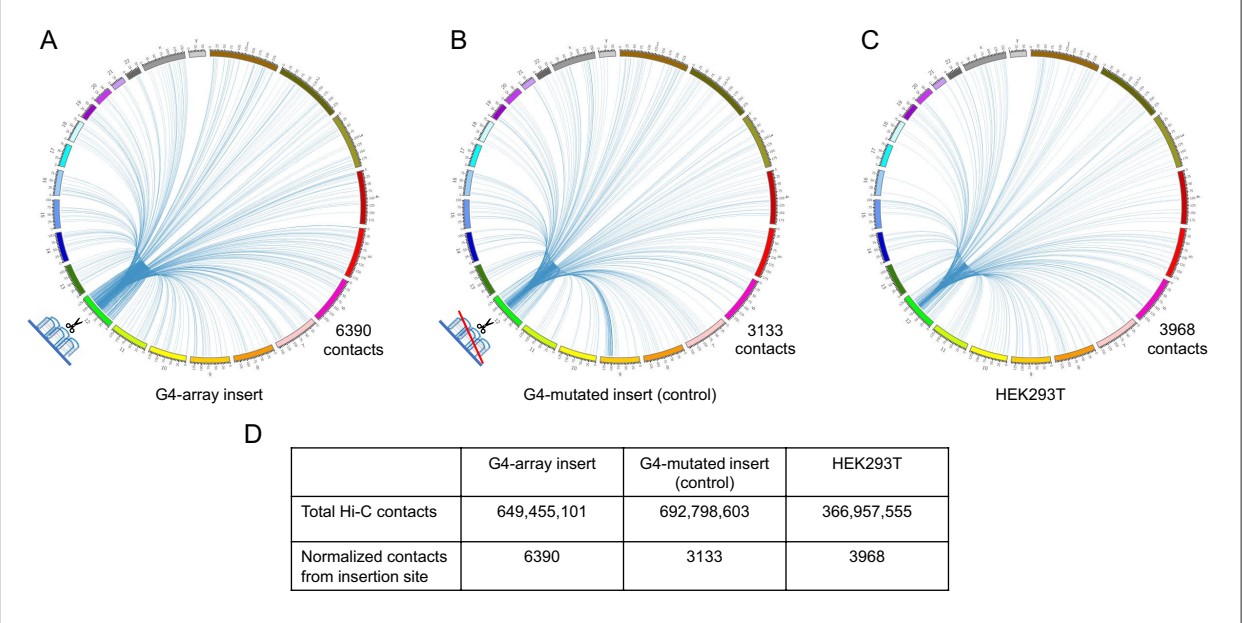

**Figure 5.** Insertion of the G4-array increased Hi-C interactions. Circos plots showing raw Hi-C contacts across the genome originating from a±10 kb window with the insertion site at the middle across three samples- (**A**) G4-array insert cells, (**B**) G4-mutated insert (control) cells and (**C**) HEK293T control cells (taken from GSE44267). (**D**) Table showing the number of genome-wide raw Hi-C contacts and normalized contacts (normalized against the total raw Hi-C contacts to normalize for the sequencing depth) originating from the ±10 kb window with the insertion site at the middle across the three samples.

long-range interactions were largely within the TAD, and limited in the upstream region due to the TAD boundary. This is clearly seen in *Figure 6C*, akin to an 'architectural stripe' displaying that the inserted G4 array had enhanced Hi-C interactions across the domain, thus prominently featured in the downstream regions.

## G4-array insertion at a second locus gives enhancer-like functions

Finally, we checked if enhancer-like effects were observed upon insertion of G4 array at another locus. Like the first site of insertion, we first identified an isolated locus devoid of G4s in the vicinity and with low interactions with surrounding regions near the 10 millionth position of chromosome 12 (10M hereafter, chr12:10588429–10588430, hg19; *Figure 7—figure supplement 1*). The G4-array, or its G4-mutated (control) sequences were inserted at the 10 M locus (*Figure 7A and B*).

As for the 79 M locus, to validate intracellular G4 formation and study the chromatin state at the inserted locus, PCR primers were designed keeping multiple points in mind (as described above). Here, for testing formation of G4 at the 10 M insertion, we used primer pairs R2 (scheme for 10 M shown in *Figure 7* top panel), covering >100 bases of the inserted G4-array, or the G4-mutated control. BG4 ChIP-qPCR validated formation of intracellular G4s within the G4 array, and not the G4-mutated control sequence (*Figure 7—figure supplement 2*). Next, we checked for changes in chromatin and the surrounding gene expression due to G4 formation. A relative increase in the H3K4Me1 and H3K27Ac enhancer marks in the G4-array was evident compared to the G4-mutated control (*Figure 7C and D*), consistent with earlier observations following G4 insertion at the 79 M locus (*Figure 2A and B*). We noticed, however, that the enhanced levels of H3K27Ac were not as marked as the 79 M locus. On the other hand, interestingly, relative increase in the H3K27Me3 repressor mark compared to the control mutated-G4 insert, particularly at the downstream end of the insertion locus was seen (*Figure 7E*). There was no G4-specific change in the presence of the chromatin compaction mark H3K9Me3, or the promoter activation mark H3K4Me3 (*Figure 7F and G*). As expected from earlier observations and the enhancer histone marks, there was a G4-dependent increase in the expression of surrounding genes *KLRC2*, *KLRC1* and *NTF3*; except for *PTPRO*, which had reduced expression (*Figure 7H*). Taken together, G4-specific chromatin changes were evident at the 10 M locus consistent with the 79 M locus. Notable variations however must be pointed out: like

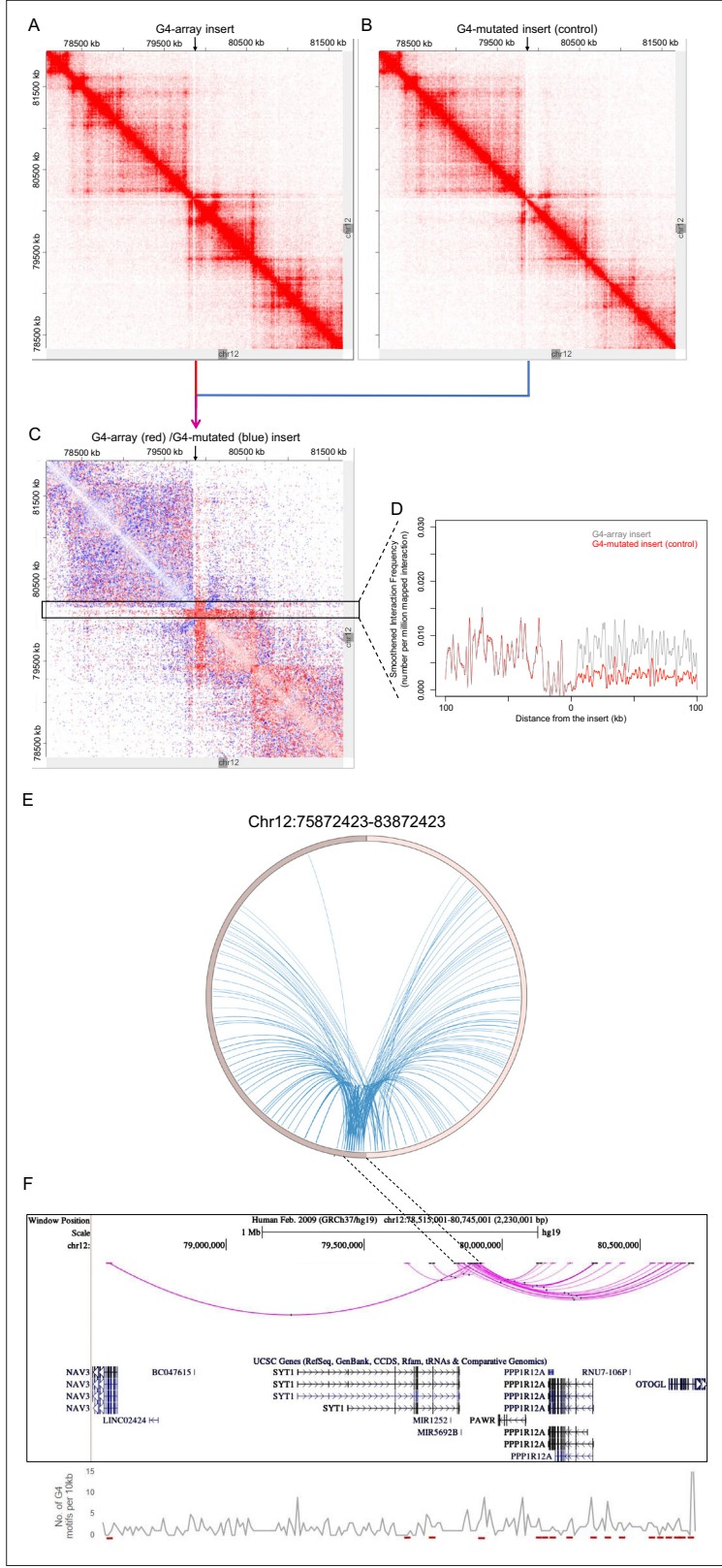

**Figure 6.** G4-dependent changes in local chromatin architecture. Juicebox Hi-C matrices showing Hi-C contacts in the (**A**) G4-array insert cells, (**B**) G4-mutated insert (control) cells in a 3.6 Mb region of chromosome 12 with the insertion site at the middle of the matrices. The arrows at the top of the Hi-C matrices indicate the site of insertion. (**C**) Juicebox Hi-C matrix showing normalized Hi-C contacts in the G4-array insert cells over the G4-mutated

*Figure 6 continued on next page*

*Figure 6 continued*

insert (control) cells as a heatmap. The region of interest (i.e. interactions associated with the immediate vicinity of the insert) is marked with a box. The arrow at the top of the Hi-C matrix indicates the site of insertion. (**D**) A line histogram displaying the differences in interaction frequency across G4-array insert cells and G4-mutated insert (control) cells in regions up to 100 kb away from the insertion site. As seen interactions downstream of the insertion site are more enriched than upstream in the G4-array insert cells as compared to the G4-mutated control. (**E**) Circos plot showing differential interactions (fold enrichment ≥ 2) originating from a±100 kb window with the insertion site at the middle, in the G4-array insert cells over the G4-mutated insert (control) cells. (**F**) UCSC genome browser snapshot showing the more significant differential interactions (fold enrichment ≥ 2, interaction reads >20) originating from a±50 kb window with the insertion site at the middle, in the G4-array insert cells over the G4-mutated insert (control) cells. The color intensity of the arcs indicating the interacting bins is proportional to the fold enrichment. Density of potential G4 motifs (per 10 kb) shown in lower panel; G4-forming sequences identified using pqsfinder (*Hon et al., 2017*); interaction regions marked in red at the bottom of lower panel.

The online version of this article includes the following source data and figure supplement(s) for figure 6:

**Source data 1.** Source data for *Figure 6D* (G4-dependent changes in local chromatin architecture- Interaction frequency histogram).

**Source data 2.** Source data for *Figure 6E* (G4-dependent changes in local chromatin architecture- Circos plot showing differential interactions).

**Source data 3.** Source data for *Figure 6F* (G4-dependent changes in local chromatin architecture- UCSC genome browser snapshot showing the more significant differential interactions).

**Figure supplement 1.** The chromosomal architecture of the insertion locus in the G4-array insert cells is broadly similar to uninserted cells except for the increase in looping interactions.

---

the presence of the H3K27Me3 repressor histone mark, along with H3K27Ac/H3K4Me1 enhancer histone marks, indicating a poised enhancer-like state as described earlier (*Calo and Wysocka, 2013*). These suggest the impact of G4 formation on chromatin is likely context-specific, that is, dependent on the chromatin state of the adjacent regions.

## Discussion

To directly test if G4s affect long-range chromatin organization we artificially inserted an array of G4s in the chromatin. Hi-C experiments clearly showed an enhanced number of cis- and trans-chromosomal long-range interactions emanating from the introduced G4s. This was G4-specific because a similar sequence devoid of G4-forming capability introduced at the same site did not result in enhanced interactions. Furthermore, interestingly, most new long-range interactions following G4 incorporation were downstream from the site of insertion. This is likely because the G4 insertion locus was proximal to the upstream TAD boundary thereby restricting most new interactions to the downstream regions within the TAD (*Figures 5 and 6*).

The insertion of the G4 array led to enhanced expression of genes up to 5 Mb away compared to cells with the G4 mutated control insertion. Furthermore, there was enrichment in the H3K4Me1 and H3K27Ac enhancer histone marks, along with recruitment of transcriptional coactivator p300 and more prominently the functionally active acetylated p300/CBP. This was clearly due to the introduction of G4s and not found upon the introduction of the G4-mutated control sequence. The enhancer marks were relatively reduced, although not markedly, when the inserted G4s were specifically disrupted (*Figure 4*).

The enhancer-like effects observed upon G4 insertion at the 79 M locus were largely replicated when the same G4-array was inserted in another locus (10 M). Interestingly, the G4-dependent increase in H3K27Me3 at the 10 M locus, in addition to the enhancer mark H3K4Me1, supports formation of a poised enhacer-like state, as described earlier (*Calo and Wysocka, 2013*). Although both the inserted G4 regions induced enhancer-like chromatin, notable context-specific influence, likely due to the chromatin-state at the regions adjacent to the insertion locus, were evident. Taken together, findings here directly support the function of G4s as enhancer-like elements and as factors that enhance long-range chromatin interactions. It is possible that such interactions are also contextually dependent on the type of G4 structure, in addition to the adjacent sequence context, and further studies will be necessary to elucidate these.

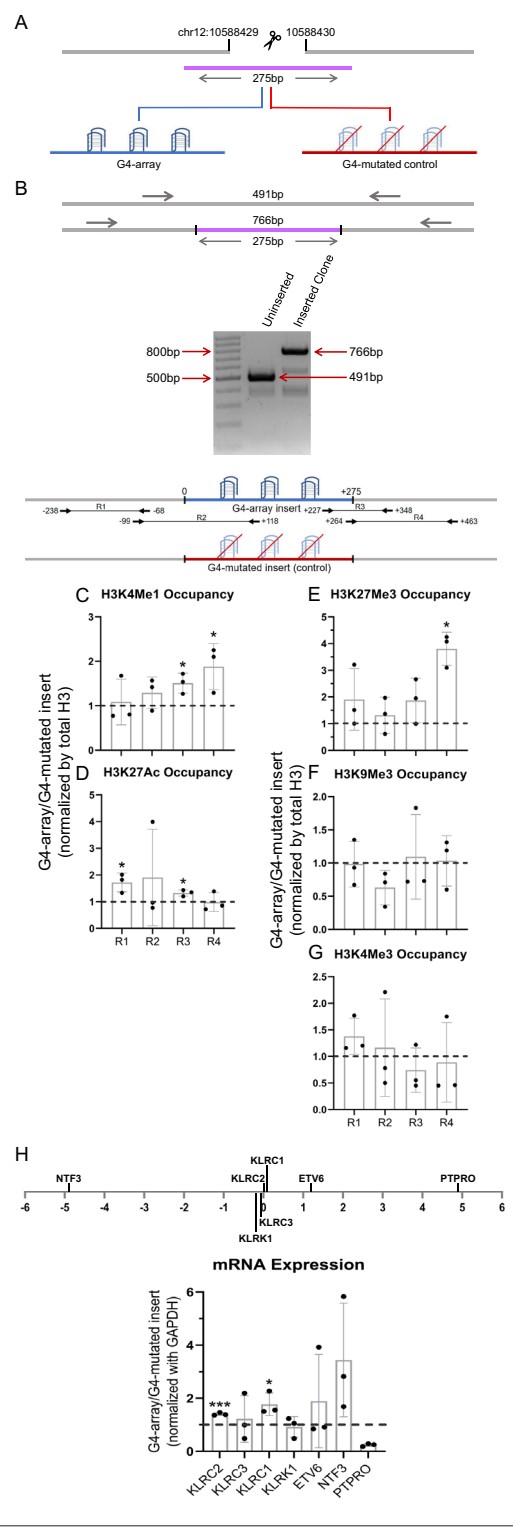

**Figure 7.** Insertion of the G4-array in another isolated locus and subsequent changes in chromatin and surrounding gene expression. (**A**) Schematic showing the insertion of the G4-array and the G4-mutated control at chr12:10,588,429–10,588,430 (hg19). (**B**) PCR of the insertion locus showing the successful insertion

*Figure 7 continued*

of the 275 bp long insert sequence. The top panel shows the positions of the PCR amplicons used in the Histone ChIP experiments. Changes in chromatin-modifying histone modifications in the insert region represented by calculating the ratio of occupancy of different histone marks in the G4-array insert cells over the G4-mutated insert (control) cells- enhancer mark, H3K4Me1 (**C**); active enhancer/promoter mark, H3K27Ac (**D**); facultative repressor mark, H3K27Me3 (**E**); constitutive repressor mark, H3K9Me3 (**F**) and active promoter mark, H3K4Me3 (**G**). (**H**) Long-range G4-dependent changes in mRNA expression are represented by calculating the ratio of expression of surrounding genes in the G4-array insert cells over the G4-mutated insert (control) cells. The panel above shows the positions of the gene promoters with respect to the insertion site. Mean ± SD (n=3); unpaired, two-tailed t-test (*p<0.05, **p<0.01, ***p<0.001, ****p<0.0001).

The online version of this article includes the following source data and figure supplement(s) for figure 7:

**Source data 1.** Source data for *Figure 7* (Insertion of the G4-array in another isolated locus and subsequent changes in chromatin and surrounding gene expression).

**Source data 2.** Raw blot shown in *Figure 7B*.

**Figure supplement 1.** Another isolated locus was chosen for insertion.

**Figure supplement 2.** G4 formation analyzed by BG4 ChIP.

**Figure supplement 2—source data 1.** Source data for *Figure 7—figure supplement 2* (G4 formation analyzed by BG4 ChIP).

---

To ensure that the observed effects were from intracellular G4 formation, we accounted for the following while designing the experiments. First, we introduced an array of G4s and confirmed in vivo G4 formation; the inserted sequence was from the hTERT promoter region with multiple arrayed G4s (*Lim et al., 2010*; *Monsen et al., 2020*; *Palumbo et al., 2009*). Second, we selected an insertion locus that was otherwise devoid of intrinsic G4s in a±2.5 kb window. Third, the selected insertion locus was relatively sparse in long-range interactions. Fourth, we independently inserted a sequence of identical length (and similar GC%) which does not form G4s at the same locus (G4-mutated control). All results were compared to the G4-mutated inser-tion. Although the introduced mutations for the control sequences were minimal, the impacts of such mutations on the binding of specific

transcription factors associated with the sequence, particularly SP1, reported to bind G4s (*Raiber et al., 2012*), cannot be ruled out.

Existing literature shows promoter G4s are involved in regulating gene expression (*Huppert and Balasubramanian, 2007*; *Rawal et al., 2006*; *Verma et al., 2008*). Additionally, G4s have been reported to regulate chromatin epigenetics through both cytosine methylation and histone modifications (*Halder et al., 2010*; *Mao et al., 2018*; *Sengupta et al., 2021*). Previous studies by us further show that promoter G4s regulate gene expression by recruiting histone-modifying regulatory complexes (*Hussain et al., 2017*; *Mukherjee et al., 2019*; *Saha et al., 2017*; *Sharma et al., 2021*). Here, we aimed to study how G4s affect the expression of genes far from their location, and if this was through G4-induced modifications in long-range 3D chromatin interactions.

Multiple studies have correlated the presence of G4s with long-range associations. CTCF, an architectural protein primarily involved in TAD boundary formation, was observed to bind to G4s and G4 stabilization was noted to enhance CTCF occupancy (*Tikhonova et al., 2021*). In addition, G4s were noted to be enriched in TAD boundaries and associated with the formation of chromatin loops (*Hou et al., 2019*). G4s were also found to coincide with open chromatin regions and H3K27Ac and H3K4Me1 ChIP-Seq peaks, which are markers for transcriptional enhancers (*Calo and Wysocka, 2013*; *Hou et al., 2021*; *Lyu et al., 2022*; *Shlyueva et al., 2014*). Most of these regions were observed to overlap with annotated enhancers and promoters regulated by such enhancers were enriched in G4s (*Williams et al., 2020*). A recent G4 CUT&Tag study further noted G4 formation at both active promoters and active and poised enhancers (*Lyu et al., 2022*).

Further, it was proposed that inter-molecular G4 formation between distant stretches of Gs may lead to DNA looping (*Hegyi, 2015*; *Liano et al., 2022*). Consistent with this, we noted with interest promoters of the four genes (*PAWR, PPP1R12A, NAV3* and *SLC6A15*; *Figure 3A*), activated on long-range interaction with the inserted loci, harbor potential G4-forming sequences (pG4) (*Figure 3—figure supplement 1*). Further, we analyzed the long-range contact regions shown in *Figure 6F*, along with the whole locus, for pG4s (*Hon et al., 2017*). Relative enrichment in pG4s was evident, particularly within the significantly enhanced contact points, at times spreading beyond the interacting region (*Figure 6F*, lower panel). Together, these support G4-induced long-range interactions.

The YY1 transcription factor was found to bind to G4s and dimerization of G4-bound YY1 led to chromatin looping interactions and consequent regulation of target gene expression (*Li et al., 2021*). A recent study also showed R-loops and possibly R-loop-associated G4 formation are enriched at CTCF binding sites, and stronger CTCF binding facilitated by G4s promotes chromatin looping (*Wulfridge et al., 2023*). In addition, it was shown that G4s assist in RNA polymerase II-associated chromatin looping (*Yuan et al., 2023*). In this context, further work will be required to understand whether and how formation of R-loops or RNA-DNA hybrid G4s (*Fay et al., 2017*), and/or association of factors like cohesion and CTCF, at the G4-array insertion sites impact chromatin looping.

In summary, our findings here demonstrate a causal role of G4s in inducing both long-range associations and enhancer function. Findings from a G4-forming stretch inserted at two independent loci illustrate the function of G4s in 3D gene regulation. Together these shed new mechanistic light on how DNA secondary structure motifs directly control the state of 3D chromatin and thereby biological function.

## Materials and methods

### Key resources table

| Reagent type (species) or resource | Designation | Source or reference | Identifiers | Additional information |
|---|---|---|---|---|
| Cell line (*H. sapiens*) | HEK 293T | NCCS Cell Repository | RRID:CVCL_0063 | |
| Transfected construct (*S. pyogenes*) | pX459 v2.0 | Addgene | #62988 | Construct to co-express cas9 protein and the gRNAs |
| Antibody | Histone H3 rabbit polyclonal | Abcam | ab1791, RRID:AB_302613 | (5 µg) |
| Antibody | H3K4Me1 rabbit polyclonal | Abcam | ab8895, RRID:AB_306847 | (5 µg) |
| Antibody | H3K27Ac rabbit polyclonal | Abcam | ab4729, RRID:AB_2118291 | (5 µg) |
| Antibody | H3K4Me3 mouse monoclonal | Abcam | ab1012, RRID:AB_442796 | (5 µg) |

*Continued on next page*

*Continued*

| Reagent type (species) or resource | Designation | Source or reference | Identifiers | Additional information |
|---|---|---|---|---|
| Antibody | H3K27Me3 mouse monoclonal | Abcam | ab6002, RRID:AB_305237 | (5 µg) |
| Antibody | H3K9Me3 rabbit polyclonal | Abcam | ab8898, RRID:AB_306848 | (5 µg) |
| Antibody | p300 rabbit monoclonal | CST | #54062, RRID:AB_2799450 | (5 µg) |
| Antibody | Ac-p300/CBP rabbit polyclonal | CST | #4771, RRID:AB_2262406 | (5 µg) |
| Antibody | BG4 antibody | Sigma-Aldrich | MABE917, RRID:AB_2750936 | (5 µg) |
| Commercial assay or kit | Arima-HiC Kit | Arima Genomics | A510008 | |
| Software, algorithm | Juicer | https://github.com/aidenlab/juicer | | |
| Software, algorithm | Juicebox | https://github.com/aidenlab/Juicebox/wiki/Download | | |
| Software, algorithm | Bedtools | https://bedtools.readthedocs.io/en/latest/ | | |
| Software, algorithm | HOMER | http://homer.ucsd.edu/homer/ | | |

## Cell lines and cell culture conditions

HEK293T cells were procured from the NCCS cell repository, the cell identity was authenticated using STR profiling, and the cells were tested negative for mycoplasma contamination. The cells were cultured in Dulbecco's Modified Eagle's Medium- High Glucose (DMEM-HG) supplemented with 10% FBS and 1XAnti-Anti (Gibco).

## Primary antibodies

Histone H3 rabbit polyclonal (Abcam ab1791), H3K4Me1 rabbit polyclonal (Abcam ab8895), H3K27Ac rabbit polyclonal (Abcam ab4729), H3K4Me3 mouse monoclonal (Abcam ab1012), H3K27Me3 mouse monoclonal (Abcam ab6002), H3K9Me3 rabbit polyclonal (Abcam ab8898), p300 rabbit monoclonal (CST 54062), Ac-p300/CBP rabbit polyclonal (CST 4771), BG4 antibody (Sigma-Aldrich MABE917).

## Genomic insertions using CRISPR-Cas9 genome editing

For the genomic insertions CRISPR-Cas9 genome editing technique was used (*Ran et al., 2013*). For the G4 array insertion, 275 bp long *hTERT* promoter region was PCR amplified from HEK 293T genomic DNA. For the insertion of the mutated G4s, a synthetic DNA template was synthesized and cloned into pUC57 vector by Genscript Biotech Corp, where 12 Gs were substituted with Ts (see *Supplementary file 2* for detailed sequences). Both the G4 array and the G4 mutated insertion templates were PCR amplified using longer primers where the short homology arms were introduced as overhangs of the primer for the accurate insertion at the 79 M locus via homologous recombination (see *Supplementary file 2* for primer sequences) (*Paix et al., 2017*). For cleavage at the 79 M locus (chr12:79,872,423–79,872,424 (hg19)), the gRNA sequence, 5'-ACTATGTATGTACATCCAGG-3', was cloned into the pX459 v2.0, a gift from Feng Zhang, that co-expresses cas9 protein and the gRNA. For cleavage at the 10 M locus (chr12:10,588,429–10,588,430 (hg19)), the gRNA sequence, 5'-ATCC TTCCCTGAATCATCAA-3', was used. Guide RNAs (gRNAs) were designed using the CRISPOR tool (*Haeussler et al., 2016*). Once the gRNA cloned vector and the insertion donor templates were ready, they were transfected into HEK293T cells and the transfected cells were selected using puro-mycin, whose resistance gene was present in the pX459 vector. Then these selected cells were serially diluted to isolate clones originating from single cells. Many such clones were screened to detect cells with homozygous/heterozygous insertion of the G4 array or mutated G4 insert by performing locus-specific PCR. Either primers adjacent to the insertion site or cross primers, i.e., one primer within the insert and another from the adjacent region, were used to screen and identify insertions. While using adjacent primers, a shift in PCR product with an increase in amplicon size by 275 bp (size of the insert) indicated successful insertion (see *Supplementary file 2* for primer sequences).

## ChIP (chromatin immunoprecipitation)

ChIP assays were performed as per the protocol previously reported in *Mukherjee et al., 2018*. Immunoprecipitation was done using relevant primary antibodies. IgG was used for isotype control.

Total histone H3 was used as a control for the histone modifications ChIP. Three million cells were harvested and crosslinked with ~1% formaldehyde for 10 min and lysed. Chromatin was sheared to an average size of ~250–500 bp using Biorupter (Diagenode). Ten percent of sonicated fraction was processed as input using phenol–chloroform and ethanol precipitation. ChIP was performed using 3 µg of the respective antibody incubated overnight at 4 °C. Immune complexes were collected using salmon sperm DNA-saturated magnetic protein G Dynabeads (Anti-FLAG M2 magnetic beads for BG4 ChIP) and washed extensively using a series of low salt, high salt and LiCl Buffers. The Dynabeads were then resuspended in TE (Tris- EDTA pH 8.1) buffer and treated with proteinase K at 65 ° C for ~5 hrs. Then, phenol-chloroform-isoamyl alcohol was utilized to extract DNA. Extracted DNA was precipitated by centrifugation after incubating overnight at –20 ° C with isopropanol, 0.3 M sodium acetate and glycogen. The precipitated pellet was washed with freshly prepared 70% ethanol and resuspended in TE buffer. ChIP DNA was analyzed by qRT-PCR method. See *Supplementary file 2* for primer sequences.

## Real-time PCR for gene (mRNA) expression

Total RNA was isolated using TRIzol Reagent (Invitrogen, Life Technologies) according to the manufacturer's instructions. RNA was quantified and cDNA was synthesized using iScript cDNA Synthesis Kits. A relative transcript expression level for genes was measured by quantitative real-time PCR using a SYBR Green based method (see *Supplementary file 2* for primer sequences). Average fold change was calculated by the difference in threshold cycles (Ct) between test and control samples. GAPDH gene was used as internal control for normalizing the cDNA concentration of each sample.

## Chromosome conformation capture (3C)

Chromosome Conformation Capture (3 C) assay was done as per the protocol reported in *Cope and Fraser, 2009* with certain modifications. Briefly, about 5–6 million cells were crosslinked using 1% formaldehyde for 10 min and then lysed to isolate the nuclei. Nuclei were digested overnight by HindIII and then ligated in a diluted reaction so that intramolecular ligation is favored. After ligation, the reaction mixture was treated with proteinase K at 65 °C to de-crosslink the DNA, followed by RNase A treatment. Then, phenol-chloroform-isoamyl alcohol was utilized to extract DNA. Extracted DNA was precipitated by centrifugation after incubating overnight at –80 °C with 70% ethanol, 0.1 M sodium acetate and glycogen. The precipitated pellet was washed with freshly prepared 70% ethanol and resuspended in TE buffer. 3C looping interactions were analyzed by TaqMan qRT-PCR method. For comparison, each interaction frequency was normalized to the interaction between exons 2 and 8 of the human α-actin (ACTA2)(*Hadjur et al., 2009*). See *Supplementary file 2* for primer sequences.

## G4 disruption using LNA probes

Probes were designed to specifically bind to regions of genomic DNA containing G repeats which would form the G stems of the G4 structure. The probes containing LNA nucleotides should hybridize with the target with higher stability than the stability of the G4 structure thus destabilizing the G4. The probes used to target the G4 array insert were: 5'-**c\*cc**ga**cccc**tcc**\*c**-3', 5'-**c\*c**ag**cccc**tcc\***g**-3', 5'-**c\*ccc**t**cccc**ttc\*c-3'. Stretches of three or more Cs are shown in bold, LNA nucleotides within the probes are underlined, the ends of the probes were protected using phosphorothioate bonds, shown as \*. Approximately 0.8 µg of LNA probes (all three mixed in equimolar amounts) were transfected per million cells. Cells were treated with the LNA probes for 108 hr by transfecting thrice with a gap of 36 hr in between. *Scheme 1* shows the LNA probes designed to disrupt the inserted G4 structures along with the inserted G4 array sequence to show the specific sites of hybridization by the LNA probes.

CCAGGCC**GGG**CT**CCC**AGTGGATTCGC**GGG**CACAGACG**CCC**AGGACCGCGCTT**CCC**ACGTGGCG
GA**GGG**ACT**GGGG**A**CCC**GGGCA**CCC**GTCCTG**CCCC**TTCACCTTCCAGCTCCGCCTCCTCCGCGCG
GA**CCCC**GC**CCC**GT**CCC**GA**CCCC**TCCCGGG**TCCCC**GGC**CCAG**GC**CCCC**TCCGGGGCCCTCCCAG**CC**
**CCT**CCCCTTCCTTTCCGCGG**CCCC**G**CCC**TCTCCTCGCGGCGCGAGTTTCAGGCAGCGCTGCGTC
CTGCTGCGCACGT**GGG**AAGCC

C*CCGACCCCTCC*C

C*CAGCCCCCTCC*G

C*CCCTCCCCTTC*C

**Scheme 1.** LNA probes designed to disrupt the inserted G4 structures along with the inserted G4 array sequence to show the specific sites of hybridization by the LNA probes.

## Hi-C

Hi-C was performed using the Arima-HiC Kit as per the manufacturer's protocol. After the proximally-ligated Hi-C templates were generated, sequencing libraries were prepared using NEBNext Ultra II DNA Library Prep Kit as per the Arima-HiC Kit's protocol. The quality of the sequencing libraries was cross-checked using TapeStation (Agilent Technologies) and the KAPA Library Quantification Kit (Roche) before proceeding with sequencing using NovaSeq 6000 (Illumina).

## Hi-C data analysis

Hi-C reads were mapped to the hg19 human genome and processed using default parameters using Juicer (https://github.com/aidenlab/juicer, **Aiden Lab, 2023a**; **Durand et al., 2016b**). Hi-C count matrices were generated at 5 kb, 10 kb, 25 kb, 50 kb, 100 kb, and 250 kb using Juicer. Hi-C heatmap figures were rendered using Juicebox (https://github.com/aidenlab/Juicebox/wiki/Download, **Aiden Lab, 2023b**; **Durand et al., 2016a**). Hi-C contacts originating in the loci flanking the G4 insertion site were generated using bedtools (https://bedtools.readthedocs.io/en/latest/, **Quinlan, 2023**). The circos plots were rendered using Circos (https://circos.ca/, **Krzywinski, 2009**). To identify significant interaction the data was processed using HOMER (http://homer.ucsd.edu/homer/, **Benner, 2024**) using analyzeHiC function. The bins showing 2-fold enrichment in G4 WT over G4 Mut and vice-versa were retained for filtering contacts for representation on circos plots.

## Materials availability

HEK 293T cells with the hTERT promoter G4 array or the G4-mutated control insertions are available upon request. Such requests can be directed to the corresponding author.

## Acknowledgements

This work was supported by the DBT/Wellcome Trust India Alliance (IA/S/18/2/504021).

## Additional information

### Funding

| Funder | Grant reference number | Author |
|---|---|---|
| DBT/Wellcome Trust India Alliance | IA/S/18/2/504021 | Shantanu Chowdhury |

The funders had no role in study design, data collection and interpretation, or the decision to submit the work for publication.

### Author contributions

Shuvra Shekhar Roy, Resources, Data curation, Formal analysis, Investigation, Visualization, Methodology, Writing - original draft; Sulochana Bagri, Resources, Data curation, Investigation; Soujanya Vinayagamurthy, Claudia Regina Then, Investigation; Avik Sengupta, Visualization; Rahul Kumar, Formal analysis, Visualization; Sriram Sridharan, Formal analysis, Visualization, Methodology; Shantanu

Chowdhury, Conceptualization, Supervision, Funding acquisition, Methodology, Project administration, Writing - review and editing

## Author ORCIDs
Shuvra Shekhar Roy ⬦ https://orcid.org/0000-0001-6005-2767
Sulochana Bagri ⬦ http://orcid.org/0000-0001-7407-2558
Soujanya Vinayagamurthy ⬦ https://orcid.org/0000-0003-1465-9925
Rahul Kumar ⬦ https://orcid.org/0000-0002-6927-5390
Shantanu Chowdhury ⬦ https://orcid.org/0000-0001-7185-8408

Reviewer #1 (Public review): https://doi.org/10.7554/eLife.96216.3.sa1
Reviewer #2 (Public review): https://doi.org/10.7554/eLife.96216.3.sa2
Reviewer #3 (Public review): https://doi.org/10.7554/eLife.96216.3.sa3
Author response https://doi.org/10.7554/eLife.96216.3.sa4

# Additional files

## Supplementary files
• Supplementary file 1. Comparative analysis of the G4-dependent increase in Hi-C interactions upon insertion. Table showing the number of genome-wide raw Hi-C contacts, actual number of contacts originating from the ±10 kb window with the insertion site at the middle and the comparative analysis (mean, standard deviation and z-score) of these number of contacts with contacts across 10,000 random 20 kb windows across the genome across the 3 samples.

• Supplementary file 2. Supplementary materials and methods.

• MDAR checklist

## Data availability
The sequencing data underlying this article are available in the NCBI Sequence Read Archive, accessible using the following link- https://www.ncbi.nlm.nih.gov/bioproject/PRJNA1048044. The rest of the data are available within the manuscript and supporting files. Further inquiries can be directed to the corresponding author.

The following dataset was generated:

| Author(s) | Year | Dataset title | Dataset URL | Database and Identifier |
|---|---|---|---|---|
| Roy SS, Bagri S, Vinayagamurthy S, Sengup A, Then CR, Kumar R, Chowdhury S | 2023 | Inserted G-quadruplexes enhance chromosomal looping interactions | https://www.ncbi.nlm.nih.gov/bioproject/PRJNA1048044 | NCBI BioProject, PRJNA1048044 |

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
