## [Editor Report · eLife assessment]

This **valuable** study demonstrates that genomic insertion of a G4-containing sequence can be sufficient to induce chromosome loops and alter gene expression. The evidence supporting the conclusions is **convincing**. Effects were shown by Hi-C as well as qPCR for chromatin modifications and expression, and the specificity of the effects was controlled by mutating the G4-containing sequence or treating with LNA probes to abolish G4 structure formation. The work will be of interest to researchers working on chromatin organization and gene regulation.

---

## [Referee Report · Reviewer #1 (Public review)]

In this manuscript, Chowdhury and co-workers provide interesting data to support the role of G4-structures in promoting chromatin looping and long-range DNA interactions. The authors achieve this by artificially inserting a G4-containing sequence in an isolated region of the genome using CRISPR-Cas9 and comparing it to a control sequence that does not contain G4 structures. Based on the data provided, the authors can conclude that G4-insertion promotes long-range interactions (measured by Hi-C) and affects gene expression (measured by qPCR) as well as chromatin remodelling (measured by ChIP of specific histone markers).

In this revised version of the manuscript, G4 formation of the inserted sequence was validated by ChIP-qPCR, and the same G4-containing sequence was inserted at a second locus, and similar, though not identical, effects on chromatin and gene expression were observed.

Strengths:

This is the first attempt to connect genomics datasets of G4s and HiC with gene expression.

The use of Cas9 to artificially insert a G4 is also very elegant.

---

## [Referee Report · Reviewer #2 (Public review)]

Roy et al. investigated the role of non-canonical DNA structures called G-quadruplexes (G4s) in long-range chromatin interactions and gene regulation. Introducing a G4 array into chromatin significantly increased the number of long-range interactions, both within the same chromosome (cis) and between different chromosomes (trans). G4s functioned as enhancer elements, recruiting p300 and boosting gene expression even 5 megabases away. The study reveals that G4s directly influence 3D chromatin organization via facilitating communication between regulatory elements and genes.

Strengths:

The authors' findings are valuable for understanding the role of G4-DNA in 3D genome organization and gene transcription. The authors provide convincing evidence to support their claims.

---

## [Referee Report · Reviewer #3 (Public review)]

Summary:

This paper aims to demonstrate the role of G-quadruplex DNA structures in the establishment of chromosome loops. The authors introduced an array of G4s spanning 275 bp, naturally found within a very well characterized promoter region of the hTERT promoter, in an ectopic region devoid of G-quadruplex and annotated gene. As a negative control, they used a mutant version of the same sequence in which G4 folding is impaired. Due to the complexity of the region, 3 G4s on the same strand and one on the opposite strand, 12 point mutations were made simultaneously (G to T and C to A). Analysis of the 3D genome organization shows that the WT array establishes more contact within the TAD and throughout the genome than the control array. Additionally, a slight enrichment of H3K4me1 and p300, both enhancer markers, was observed locally near the insertion site. The authors tested whether the expression of genes located either nearby or away up to 5 Mb were up-regulated based on this observation. They found that four genes were up-regulated from 1.5 to 3 fold. An increased interaction between the G4 array compared to the mutant was confirmed by the 3C assay. For in-depth analysis of the long-range changes, they also performed Hi-C experiments and showed a genome-wide increase in interactions of the WT array versus the mutated form.

Strengths:

The experiments were well-executed and the results indicate a statistical difference between the G4 array inserted cell line and the mutated modified cell line.

Weaknesses:

(1) It would have been nice to have an internal control corresponding to a region known to be folded in several cell lines to compare the level of pG4 signal within their construct with a well-characterised control (for example, the KRAS promoter region).

(2) The mutations introduced into the G4 sequence may also affect Sp1 or other transcription factor binding sites present in this region, and some of the observations may depend on these sites rather than G4 structures. While this is acknowledged in the text, the conclusion in the title of the paper seems an overstatement.

---

## [Author Response]

[The following is the authors’ response to the original reviews.]

**Reviewer #1 (Public Review):**
Summary:In this manuscript, Chowdhury and co-workers provide interesting data to support the role of G4-structures in promoting chromatin looping and long-range DNA interactions. The authors achieve this by artificially inserting a G4-containing sequence in an isolated region of the genome using CRISPR-Cas9 and comparing it to a control sequence that does not contain G4 structures. Based on the data provided, the authors can conclude that G4-insertion promotes long-range interactions (measured by Hi-C) and affects gene expression (measured by qPCR) as well as chromatin remodelling (measured by ChIP of specific histone markers).Whilst the data presented is promising and partially supports the authors' conclusion, this reviewer feels that some key controls are missing to fully support the narrative. Specifically, validation of actual G4-formation in chromatin by ChIP-qPCR (at least) is essential to support the association between G4-formation and looping. Moreover, this study is limited to a genomic location and an individual G4-sequence used, so the findings reported cannot yet be considered to reflect a general mechanism/effect of G4-formation in chromatin looping.Strengths:This is the first attempt to connect genomics datasets of G4s and HiC with gene expression. The use of Cas9 to artificially insert a G4 is also very elegant.Weaknesses:Lack of controls, especially to validate G4-formation after insertion with Cas9. The work is limited to a single G4-sequence and a single G4-site, which limits the generalisation of the findings.

In the revised version we validated G4 formation inside cells at the insertion site using the reported G4-selective antibody BG4. Significant BG4 binding (by ChIP-qPCR) was clear in the G4-array insert, and not in the G4-mutated insert, supporting formation of G4s by the inserted G4-array (included as Figure S4).

To directly address the second point, we inserted the G4-sequence, or the mutated control, at a second relatively isolated locus (at the 10 millionth position on Chr12, denoted as 10M site in text). First, BG4 ChIP was done to confirm intracellular G4 formation by the inserted array. BG4 ChIP-qPCR binding was significant within the inserted region, and not in the negative control region (Figure S8), consistent with the 79M locus. Together these demonstrate intracellular G4 formation by inserted sequences at two different loci.

We next checked the state of chromatin of the G4-array inserted at the 10M locus, or its negative control. Histone marks H3K4Me1, H3K27Ac, H3K27Me3, H3K9me3 and H3K4Me3 were tested at the G4-array, or the negative control locus. Relative increase in the enhancer histone marks was evident, relative to the control sequence. This was largely similar to the 79M locus, supporting an enhancer-like state. Interestingly, here we further noted presence of the H3K27me3 histone mark. The presence of the H3K27Me3 repressor histone mark, along with H3K4Me1/H3K27Ac enhancer histone marks, support a poised enhancer-like status of the inserted G4 region, as has been observed earlier in other studies. Together, although data from the two distinct G4 insertion sites support the enhancer-like state, there are contextual differences likely due to the sequence/chromatin of the sites adjacent to the inserted sequence.

Effect of the 10M G4-insertion on activation of surrounding genes (10 Mb window), and not the G4-mutant insert, was evident for most genes. Consistent with the enhancer-like state of the G4-array insert; in line with the 79M G4-array insert.

These results have been added as the final section in the revised version, data is shown in Figure 7.

**Reviewer #2 (Public Review):**
Summary:Roy et al. investigated the role of non-canonical DNA structures called G-quadruplexes (G4s) in long-range chromatin interactions and gene regulation. Introducing a G4 array into chromatin significantly increased the number of long-range interactions, both within the same chromosome (cis) and between different chromosomes (trans). G4s functioned as enhancer elements, recruiting p300 and boosting gene expression even 5 megabases away. The study proposes a mechanism where G4s directly influence 3D chromatin organization, facilitating communication between regulatory elements and genes.Strength:The findings are valuable for understanding the role of G4-DNA in 3D genome organization and gene transcription.Weaknesses:The study would benefit from more robust and comprehensive data, which would add depth and clarity.(1) Lack of G4 Structure Confirmation: The absence of direct evidence for G4 formation within cells undermines the study's foundation. Relying solely on in vitro data and successful gene insertion is insufficient.

Using the reported G4-specific antibody, BG4, we performed BG4 ChIP-qPCR at the 79M locus. In addition, a second G4-insertion site was created and BG4 ChIP-qPCR was used to validate intracellular G4 formation. Briefed below, more details in the response above.

In the revised version we validated G4 formation inside cells at the insertion site using the reported G4-selective antibody BG4. Significant BG4 binding (by ChIP-qPCR) was clear in the G4-array insert, and not in the G4-mutated insert, supporting formation of G4s by the inserted G4-array (included as Figure S4).

Further, we inserted the G4-sequence, or the mutated control, at a second relatively isolated locus (at the 10 millionth position on Chr12, denoted as 10M site in text). First, BG4 ChIP was done to confirm intracellular G4 formation by the inserted array. BG4-ChIP-qPCR was significant within the G4-array inserted region, and not in the negative control region (Figure S8), consistent with the 79M locus. Together these demonstrate intracellular G4 formation by inserted sequences at two different loci. Added in revised text in the second and the final sections of results, data shown in Figures 7, S4 and S8.

(2) Alternative Explanations: The study does not sufficiently address alternative explanations for the observed results. The inserted sequences may not form G4s or other factors like G4-RNA hybrids may be involved.

As mentioned in response to the previous comment, we confirmed that the inserted sequence indeed forms G4s inside the cells. RNA-DNA hybrid G4s can form within R-loops with two or more tandem G-tracks (G-rich sequences) on the nascent RNA transcript as well as the non-template DNA strand (Fay et al., 2017, 28554731). A recent study has observed that R-loop-associated G4 formation can enhance chromatin looping by strengthening CTCF binding (Wulfridge et al., 2023, 37552993). As pointed out by the reviewer, the possibility of G4-RNA hybrids remains, we have mentioned this possibility for readers in the second last paragraph of the Discussion.

(3) Limited Data Depth and Clarity: ChIP-qPCR offers limited scope and considerable variation in some data makes conclusions difficult.

We noted variation with one of the primers in a few ChIP-qPCR experiments (in Figures 2 and 3D). The changes however were statistically significant across replicates, and consistent with the overall trend of the experiments (Figures 2, 3 and 4). Enhancer function, in addition to ChIP, was also confirmed using complementary assays like 3C and RNA expression.

(4) Statistical Significance and Interpretation: The study could be more careful in evaluating the statistical significance and magnitude of the effects to avoid overinterpreting the results.

We reconfirmed our statistical calculations from biological replicate experiments. We carefully looked at potential overinterpretations, and made appropriate changes in the manuscript (details of the changes given below in response to comment to authors).

**Reviewer #3 (Public Review):**
Summary:This paper aims to demonstrate the role of G-quadruplex DNA structures in the establishment of chromosome loops. The authors introduced an array of G4s spanning 275 bp, naturally found within a very well-characterized promoter region of the hTERT promoter, in an ectopic region devoid of G-quadruplex and annotated gene. As a negative control, they used a mutant version of the same sequence in which G4 folding is impaired. Due to the complexity of the region, 3 G4s on the same strand and one on the opposite strand, 12 point mutations were made simultaneously (G to T and C to A). Analysis of the 3D genome organization shows that the WT array establishes more contact within the TAD and throughout the genome than the control array. Additionally, a slight enrichment of H3K4me1 and p300, both enhancer markers, was observed locally near the insertion site. The authors tested whether the expression of genes located either nearby or up to 5 Mb away was up-regulated based on this observation. They found that four genes were up-regulated from 1.5 to 3-fold. An increased interaction between the G4 array compared to the mutant was confirmed by the 3C assay. For in-depth analysis of the long-range changes, they also performed Hi-C experiments and showed a genome-wide increase in interactions of the WT array versus the mutated form.Strengths:The experiments were well-executed and the results indicate a statistical difference between the G4 array inserted cell line and the mutated modified cell line.Weaknesses:The control non-G4 sequence contains 12 point mutations, making it difficult to draw clear conclusions. These mutations not only alter the formation of G4, but also affect at least three Sp1 binding sites that have been shown to be essential for the function of the hTERT promoter, from which the sequence is derived. The strong intermingling of G4 and Sp1 binding sites makes it impossible to determine whether all the observations made are dependent on G4 or Sp1 binding. As a control, the authors used Locked Nucleic Acid probes to prevent the formation of G4. As for mutations, these probes also interfere with two Sp1 binding sites. Therefore, using this alternative method has the same drawback as point mutations. This major issue should be discussed in the paper. It is also possible that other unidentified transcription factor binding sites are affected in the presented point mutants.

Since the sequence we used to test the effects of G4 structure formation is highly G-rich, we had to introduce at least 12 mutations to be sure that a stable G4 structure would not form in the mutated control sequence. Sp1 has been reported to bind to G4 structures (Raiber et al., 2012). Therefore, Sp1 binding is likely to be associated with the G4-dependent enhancer functions observed here. We also appreciate that apart from Sp1, other unidentified transcription factor binding sites might be affected by the mutations we introduced. We have discussed these possibilities in the fourth paragraph of the Discussion section in the revised manuscript.

**Reviewer #1 (Recommendations For The Authors):**
Whilst the data presented is promising and partially supports the authors' conclusion, this reviewer feels that some key controls are missing to fully support the narrative used. Below are my main concerns:(1) The main thing missing in the current manuscript is to validate the actual formation of G4 in chromatin context for the repeat inserted by CRISPR-Cas. Whilst I appreciate this will form promptly a G4 in vitro, to fully support the conclusions proposed the authors would need to demonstrate actual G4-formation in cells after insertion. This could be done by ChIP-qPCR using the G4-selective antibody BG4 for example. This is an essential piece of evidence to be added to link with confidence G4-formation to chromatin looping.

To address the concern regarding whether the inserted G4 sequence forms G4s in cells, as suggested, we used the G4-selective antibody BG4. PCR primers in the study were designed keeping multiple points in mind: Primers should not bind to any site of G/C alteration in the mutated control insert; either the forward/reverse primer is from the adjacent region for specificity; covers adjacent regions for studying any effects on chromatin; and, PCRs optimized keeping in mind the repeats within the inserted sequence. Given these, primer pairs R1-R4 were chosen for further work following optimizations (Figure 2, top panel). For BG4 ChIP-qPCR we used primer pairs R2, which covered >100 bases of the inserted G4-array, or the G4-mutated control. Significant BG4 binding was clear in the G4-array insert, and not in the G4-mutated insert, demonstrating formation of G4s by the inserted G4-array (Figure S4).

In response to comment #3 below, we inserted the G4-forming sequence (or its mutated control) at a second locus. This insertion was near the 10 millionth position of chromosome 12 (10M insertion locus in text). Here also, BG4 binding was significant within the G4-array inserted region, and not in the negative control region (Figure S8). Together these demonstrate G4 formation by the inserted sequence at two different loci.

(2) I found the LNA experiment very elegant. However, what would be the effect of LNA treatment on the control sequence that does not form G4s? This control is essential to disentangle the effect of LNA pairing to the sequence itself vs disrupting the G4-structure.

As per the reviewer’s suggestion, we performed a control experiment where we treated the G4-mutated insert (control) cells with the G4-disrupting LNA probes. The changes in the expression of the surrounding genes in this case were not significant, indicating that the effects observed in the G4-array insert cells were possibly due to disruption of the inserted G4 structures. This data is presented in Figure S5.

(3) The authors describe their work and present its conclusion as if this were a genome-wide study, whilst the work is focused on a specific genomic location, and the looping, along with the effect on histone acetylation and gene expression, is limited to this. The authors cannot conclude, therefore, that this is a generic effect and the discussion should be more focused on the specific G4s used and the genomic location investigated. Ideally, insertion of a different G4-forming sequence or of the same in a different genomic location is recommended to really claim a generic effect.

To address this we inserted the G4-array sequence, or the G4-mutated control sequence, at another relatively isolated locus – at the 10 millionth position of chromosome 12 – denoted as 10M. Using BG4 ChIP-qPCR intracellular G4 formation was confirmed. We observed that the enhancer-like features in terms of enhancer histone marks and increase in the expression of surrounding genes were largely reproduced at the 10M locus on G4 insertion (Figure 7). These results are added as the final section under Results.

**Reviewer #2 (Recommendations For The Authors):**
The study proposes a mechanism where G4s directly influence 3D chromatin organization, facilitating communication between regulatory elements and genes.While the present manuscript presents an interesting hypothesis, it would benefit from enhanced novelty and more robust data. The study complements existing G4 research (e.g., PMID: 31177910). While the conclusions hold biological relevance, they largely reiterate established knowledge. Furthermore, the presented data appear preliminary and still lack depth and clarity.

Hou et al., 2019 (PMID: 31177910) showed presence of potential G4-forming sequences correlated with TAD boundaries, along with enrichment of architectural proteins and transcription factor binding sites. Also, other studies noted enrichment of potential G4-forming sequences at enhancers along with nucleosome depletion and higher transcription factor binding (Hou et al., 2021; Williams et al., 2020). These studies proposed the role of G4s in chromatin/TAD states based on analysis of potential G4-forming sequences using correlative bioinformatics analyses. Here we sought to directly test causality. Insertion of G4 sequence, and formation of intracellular G4s in an isolated, G4-depleted region resulted in altered characteristics of chromatin, and not in the negative control insertion that does not form G4s. These, in contrast to earlier studies, directly demonstrates the causal role of G4s as functional elements that impact local and distant chromatin.

Major concerns:(1) Lack of G4 Structure Confirmation: Implement G4-specific antibodies or fluorescent probes to verify G4 structures inside the cells.

Detailed response given above. Briefly, in the revised version we validated G4 formation inside cells at the insertion site using the reported G4-selective antibody BG4. Significant BG4 binding (by ChIP-qPCR) was clear in the G4-array insert, and not in the G4-mutated insert, supporting formation of G4s by the inserted G4-array (included as Figure S4).

Further, we inserted the G4-sequence, or the mutated control, at a second relatively isolated locus (at the 10 millionth position on Chr12, denoted as 10M site in text). First, BG4 ChIP was done to confirm intracellular G4 formation by the inserted array. BG4 ChIP-qPCR binding was significant within the G4-array inserted region, and not in the negative control region (Figure S8), consistent with the 79M locus. Together these demonstrate intracellular G4 formation by inserted sequences at two different loci. Added in revised text in the second and the final sections of results, data shown in Figures 7, S4 and S8.

(2) Alternative Explanations: Explore the possibility that the sequences may not form G4s or that other factors like G4-RNA hybrids are involved.

Response provided in the public reviews section.

(3) Limited Data Depth and Clarity: ChIP-qPCR offers limited scope. Consider employing G4 ChIP-seq for genome-wide analysis of G4 association with histone modifications. Address inconsistencies in data like H3K27me3 variation and incomplete H3K9me3 data sets.

A recent study performed G4 CUT&Tag (Lyu et al., 2022, 34792172) and observed G4 formation at both active promoters and active and poised enhancers. We have discussed this in the sixth paragraph of the Discussion. The H3K27Me3 occupancy at the 79M locus insertions did not have any significant G4-dependent changes, however, at the second insertion site at the 10M locus (introduced in the revised manuscript, Figure 7) there was significant G4-dependent increase in H3K27Me3 occupancy along with the H3K4Me1 and H3K27Ac enhancer histone marks, indicating formation of a poised enhancer-like element.

We completed the H3K9me3 data sets for both insertion sites.

(4) Statistical Significance and Interpretation: Re-evaluate the statistical significance of results and interpret them in the context of relevant biological knowledge. Avoid overstating the impact of minor changes.

We revised several lines to avoid overstating results. Some of the changes are as below (changes underline/strikethrough)

- There was an a relatively modest increase in the recruitment of both p300 and a substantial increase in the recruitment of the more functionally active acetylated p300/CBP to the G4-array when compared against the mutated control.

- As expected, although modest, a decrease in the H3K4Me1 and H3K27Ac enhancer histone modifications was evident within the insert upon the LNAs treatment.

- Moreover, the enhancer marks were relatively reduced, although not markedly, when the inserted G4s were specifically disrupted.

(5) Unexplored Aspects: Investigate the relationship between G4 DNA and R-loops, and consider the role of CTCF and cohesin proteins in mediating long-range interactions. Integrate existing research to build a more comprehensive framework and draw more robust conclusions.

As mentioned in response to one of the earlier comments, a recent publication extensively studied the association between G4s, R-loops, and CTCF binding (Wulfridge et al., 2023). While, here we focused on the primary features of a potential enhancer, further work will be necessary to establish how G4s influence the coordinated action between cohesin and CTCF and consequent chromatin looping. We have described this for readers in the second last paragraph of the Discussion in the revised version.

Minor Concern:(1) Enhancer Definition: The term "enhancer" requires specific criteria. Modify the section heading or provide evidence demonstrating the G4 sequence fulfills all conditions for being an enhancer, such as position independence and long-range effects.

Although we checked some of the primary features of a potential enhancer: Like expression of surrounding genes, enhancer histone marks, chromosomal looping interactions, and recruitment of transcriptional coactivators, further aspects may need to be validated. As suggested, in the revised manuscript the section heading has been modified to ‘Enhancer-like features emerged upon insertion of G4s.’

**Reviewer #3 (Recommendations For The Authors):**
In addition to the points in my public review, I would like to mention some less significant points.The authors mention that "the array of G4-forming sequences used for insertion was previously reported to form stable G4s in human cells" (Lim et al., 2010; Monsen et al., 2020; Palumbo et al., 2009). However, upon reading the publications, I found that these observations were made in vitro. I may have missed something, but there are now several mappings of folded-G4 in human cells based on different approaches. It would be beneficial to investigate whether the hTERT promoter is a site of G-quadruplex formation in vivo. If confirmed, a similar analysis should be conducted on the 275 bp region inserted into the ectopic region to determine if it also has the ability to form a structured G4.

We performed BG4 ChIP to confirm in vivo G4 formation by the inserted G4-array as suggested (Figures S4, S8). Detail response given above. Briefly, in the revised version we validated G4 formation inside cells at the insertion site using the reported G4-selective antibody BG4. Significant BG4 binding (by ChIP-qPCR) was clear in the G4-array insert, and not in the G4-mutated insert, supporting formation of G4s by the inserted G4-array (included as Figure S4).

Further, we inserted the G4-sequence, or the mutated control, at a second relatively isolated locus (at the 10 millionth position on Chr12, denoted as 10M site in text). First, BG4 ChIP was done to confirm intracellular G4 formation by the inserted array. BG4-ChIP-qPCR was significant within the inserted region, and not in the negative control region (Figure S8). Consistent with the 79M locus. Together these demonstrate intracellular G4 formation by inserted sequences at two different loci. Added in revised text in the second and the final sections of results, data shown in Figures 7, S4 and S8.

The inserted sequence originates from a well-characterized promoter. The authors suggest that placing it in an ectopic position creates an enhancer-like region, based on the observation of increased levels of H3K27Ac and H3K4me1 on the WT array. To provide a control that it is not a promoter, it would be useful to also analyze a specific mark of promoter activity, such as H3K4me3.

As suggested by reviewer, we also analysed the H3K4Me3 promoter activation mark at both the 79M and 10M (introduced in the revised manuscript, Figure 7) insertion loci. We did not observe any significant G4-dependent changes in the recruitment of H3K4Me3 (Figures 2, 7).

In the discussion, the authors mention "it was proposed that inter-molecular G4 formation between distant stretches of Gs may lead to DNA looping". To investigate this further, it would be worthwhile to examine whether the promoter regions of activated genes (PAWR, PPP1R12A, NAV3, and SLC6A15) contain potentially forming G-quadruplexes (pG4). Additionally, sites that establish more contact with the G4 array described in Figure 6F could be analyzed for enrichment in pG4.

Thank you for pointing this out. We found promoters of the four genes (PAWR, PPP1R12A, NAV3, and SLC6A15) harbour potential G4-forming sequences (pG4s). Also as suggested, we analysed the contact regions in Fig 6F, along with the whole locus, for pG4s. Relative enrichment in pG4 was seen, particularly within the significantly enhanced interacting regions, which at times spreads beyond the interacting regions also. This is shown in the lower panel of Figure 6F in the revised version. We have described this in Discussion for readers.

[The following is the authors’ response to the updated reviews.]

**Reviewer #3 (Public Review):**
(2) The mutations introduced into the G4 sequence may also affect Sp1 or other transcription factor binding sites present in this region, and some of the observations may depend on these sites rather than G4 structures. While this is acknowledged in the text, the conclusion in the title of the paper seems an overstatement.

We understand the reviewer’s concern regarding the title. The title has been updated accordingly.